# Chemical Profile and Bioactivities of Extracts from Edible Plants Readily Available in Portugal

**DOI:** 10.3390/foods10030673

**Published:** 2021-03-22

**Authors:** Beatriz Nunes Silva, Vasco Cadavez, Pedro Ferreira-Santos, Maria José Alves, Isabel C. F. R. Ferreira, Lillian Barros, José António Teixeira, Ursula Gonzales-Barron

**Affiliations:** 1CEB–Centre of Biological Engineering, University of Minho, 4710–057 Braga, Portugal; beatrizsilva@ceb.uminho.pt (B.N.S.); pedrosantos@ceb.uminho.pt (P.F.-S.); jateixeira@deb.uminho.pt (J.A.T.); 2Centro de Investigação de Montanha (CIMO), Instituto Politécnico de Bragança, Campus de Santa Apolónia, 5300-253 Bragança, Portugal; vcadavez@ipb.pt (V.C.); maria.alves@ipb.pt (M.J.A.); iferreira@ipb.pt (I.C.F.R.F.); lillian@ipb.pt (L.B.)

**Keywords:** polyphenolic extracts, bioactive compounds, antioxidants, antimicrobials, preservatives, principal component analysis

## Abstract

Plant extracts have been proposed as alternative biocides and antioxidants to be included in a variety of food products. In this work, to assess the potential of French lavender, lemon balm, basil, tarragon, sage, and spearmint to be used as food additives, the chemical profiles and bioactivities of such plant extracts were studied. Furthermore, to evaluate the influence of extraction methods and solvents on the chemical characteristics and bioactivities of the plant extracts, two extraction methods (solid-liquid and Soxhlet extraction) and two solvents (water and ethanol 70% (*v*/*v*)) were tested for each plant. Groupwise summary statistics were calculated by plant, extraction method, and solvent, and linear models were built to assess the main effects of those terms and their interactions on the chemical characteristics and bioactivities of the extracts. The results revealed that all factors—type of plant, extraction method and solvent—have influence on the chemical profile and antioxidant activity of the resultant extracts. Interactions between factors were also observed. Hydroethanolic Soxhlet extracts presented the least potential as biopreservatives due to their low phenolic content and reduced antioxidant capacity. Oppositely, aqueous Soxhlet extracts and hydroethanolic solid-liquid extracts showed high contents in phenolic compounds and high antioxidant activities. In particular, the hydroethanolic solid-liquid extracts of lemon balm, spearmint, and sage presented the highest phenolic and flavonoid contents, accompanied by a high antioxidant activity, and they revealed antimicrobial activity against four pathogens (*S. enterica* ser. Typhimurium, *E. coli*, *L. monocytogenes* and *S. aureus*). These results demonstrate the potential of these natural resources to be incorporated as bioactive preservatives in foods or their packaging.

## 1. Introduction

In the food industry, synthetic preservatives have been widely used to improve or maintain the properties of foods and to extend their shelf-life. However, the safety and impact of synthetic food additives on human health has been under discussion for many years. Some studies have reported gastrointestinal, respiratory, dermatological, cardiovascular, musculoskeletal, and neurological adverse reactions, although the cause-and-effect association between food additives and symptoms is not always well documented [1].

In this sense, one of the current trends in food processing is to replace chemical additives with others that are more natural, plant-based, known to be safe from the toxicological standpoint (with GRAS status—Generally Recognized as Safe), in order to satisfy the demand of consumers for “greener” products [2]. For this reason, numerous studies on natural substances, such as plant extracts, have been conducted. These have shown promising results regarding the antimicrobial and antioxidant properties of various natural substances, thus supporting their potential as food preservatives that can be incorporated in the product or its packaging [2,3,4,5].

Nonetheless, to assure the safety of herbal extracts for human consumption, it is crucial that these originate from nontoxic solvents, such as water, ethanol, or their binary mixtures (dichloromethane, hexane, ethyl ether, chloroform, and methanol should be avoided), and from herbs with documented traditional use [6]. In this regard, there is a wide variety of plants used in traditional medicine that have been evaluated by researchers on their health-promoting, antimicrobial and antioxidant properties. This is the case of basil, lemon balm, French lavender, sage, spearmint, and tarragon [3,7,8,9,10,11], six plants that, according to recent meta-analyses, can provide protection towards pathogens in cheese [12,13].

Traditionally, the decoctions of basil (*Ocimum basilicum* L.) have been used as an herbal remedy for stomach pains, constipation, and nasal and bronchial catarrh, among other applications [3]. Moreover, basil has shown anti-inflammatory, antidiabetic, cardioprotective, immunostimulatory, anticarcinogenic and hepatoprotective properties [14].

Lemon balm (*Melissa officinalis* L.) has been widely used as a mild sedative and anxiolytic, as well as to prevent and treat gastrointestinal disorders, but other medicinal effects have also been described, including antispasmodic, antiproliferative, anti-cholinesterase and antiviral properties [15,16].

In folk medicine, French lavender (*Lavandula stoechas* L.) is a well-known aromatic plant that has been used for its anti-inflammatory, antispasmodic and carminative properties, as well as for its positive effects against various problems, including eczema, urinary tract infections and heart-burn, for example [10].

Preparations from sage (*Salvia officinalis* L.) leaf have been traditionally used in the treatment of gastrointestinal problems, and mouth and throat inflammations, for example [17]. Additionally, sage has a wide variety of pharmacological activities, such as anticancer, antimutagenic, anti-inflammatory, antinociceptive, hypoglycemic, hypolipidemic, and cognitive and memory-enhancing effects [18].

*Mentha* species, which include spearmint (*Mentha spicata* L.), have a long history of use in the treatment of respiratory problems (such as bronchitis) and digestive issues (nausea, ulcerative colitis, flatulence, etc.) [11]. The medicinal effects of *Mentha* species include anticatarrhal, anti-inflammatory, carminative, antiemetic, diaphoretic, antimutagenicity, antispasmodic, antioxidant, and analgesic activities [11].

As for tarragon (*Artemisia dracunculus* L.), this herb is commonly used in traditional medicine to treat insomnia, as a digestive stimulant, and for the treatment of skin wounds, allergic rashes, and dermatitis [19]. The main therapeutic applications reported are for the nervous, digestive and renal systems (due to the anti-epileptic, spasmolytic, laxative, and diuretic properties), for liver function, and as anti-inflammatory, anticancer and antibacterial agents [19].

With proven beneficial effects for human health—and because basil, lemon balm, French lavender, sage, spearmint, and tarragon are readily available in Portugal—further characterization of these plants was intended.

In this context, our study was designed to evaluate the chemical profiles and bioactivities of a variety of extracts, obtained from different plants, extraction techniques, and solvents. The main goals of this study were the following: (i) to deliver insight on the plant extracts with most potential to be used as food additives, among those cultivated in Portugal; and (ii) to assess the influence of distinct extraction methods and solvents on the chemical profile and antioxidant activities of plant extracts.

## 2. Materials and Methods

### 2.1. Plant Material and Extraction Procedures

Basil, lemon balm, French lavender, sage, spearmint, and tarragon dry aerial parts were provided by Pragmático Aroma Lda. (“Mais Ervas”, Trás-os-Montes, Portugal), and mechanically ground. The extractions were performed in triplicate (*n* = 3) using ethanol 70% (*v*/*v*) (Et70) and distilled water as solvents, in a shaking water bath (at 150 rpm) at 60 °C for 90 min (solid-liquid extraction); or using a Soxhlet apparatus (at 90 or 120 °C, for Et70 and distilled water, respectively), for 7 recycles (around 3.5 to 4 h). Both methods used a sample/solvent ratio of 1 g/20 mL. After filtration (filter paper of 7–10 μm), the extracts were stored in a refrigerator at 4 °C until use. For the antimicrobial assays, the extracts were frozen and lyophilized.

### 2.2. Extraction Yield

The dry weight method was used to determine the solvent efficiency in extracting compounds from the plant material. The extraction yield (presented in %) was calculated as shown in Equation (1):
(1)
$$\mathrm{Extraction}\;\mathrm{Yield}\;\left(\%\right)\;=\;\frac{\mathrm{extracted}\;\mathrm{solids}\;\left(g\right)}{\mathrm{initial}\;\mathrm{dry}\;\mathrm{material}\;\left(g\right)}\times 100$$


### 2.3. Chemical Characterisation

#### 2.3.1. Total Protein Content

The soluble protein content (TProtein) was analyzed using the Bradford assay with some modifications [20]. For this, a subsample of 20 µL plant extract was mixed with 230 µL of Bradford dye reagent. The microplate was placed in the dark for 5 min at room temperature and the absorbance was measured at a wavelength of 595 nm by an UV/Vis spectrophotometer (Synergy HT, BioTek Instruments Inc., Winooski, VT, USA). Bovine albumin serum (BSA) was used to perform the standard curve (1000–33 mg/L, *R*^2^ = 0.98) and the results were expressed as micrograms of BSA equivalents (BSAE) per gram of dry plant (μg BSAE/g dry plant).

#### 2.3.2. Total Carbohydrate Content

The carbohydrate content (Carbohyd.) was analyzed by the phenol-sulfuric acid method, as described by Masuko et al. [21]. For this, 50 µL of plant extract were mixed with 150 µL of sulfuric acid (96–98% (*v*/*v*)). Then, 30 µL of 5% phenol reagent were added and the final solution was heated for 5 min at 90 °C. After cooling at room temperature for 5 min, the absorbance was measured at 490 nm by an UV/Vis spectrophotometer (Synergy HT, BioTek Instruments Inc., Winooski, VT, USA). Glucose was used to perform the standard curve (600–10 mg/L, *R*^2^ = 0.99) and the results were expressed as micrograms of glucose equivalents (GE) per gram of dry plant (μg GE/g dry plant).

#### 2.3.3. Chlorophyll Contents

The plant extracts were analyzed for chlorophyll-a and chlorophyll-b content as described by Sumanta et al. [22]. Briefly, 2 mL of plant extract were centrifuged at 10,000 rpm for 15 min. The supernatant was collected, placed in a cuvette, and the absorbance as measured at 649 and 664 nm by an UV/Vis spectrophotometer (Synergy HT, BioTek Instruments Inc., Winooski, VT, USA). Quantification was done using Equations (2) and (3) for chlorophyll-a (Ch-a) and chlorophyll-b (Ch-b), respectively.

(2)
$$\mathrm{Ch}-a\;=\;13.36{\mathrm{Abs}}_{664\;\mathrm{nm}}-5.19{\mathrm{Abs}}_{649\;\mathrm{nm}}\;$$


(3)
$$\mathrm{Ch}-b\;=\;27.43{\mathrm{Abs}}_{664\;\mathrm{nm}}-8.12{\mathrm{Abs}}_{649\;\mathrm{nm}}$$


Results were expressed as micrograms of each photosynthetic pigment per gram of dry plant (μg Ch-a or Ch-b/g dry plant).

#### 2.3.4. Total Phenolic and Flavonoid Contents

The total phenolic content (TPC) was determined using the Folin-Ciocalteu assay [23]. For all analyses, 5 μL of plant extract (water or ethanol 70% for control) were mixed with 15 μL Folin−Ciocalteu reagent, 60 μL of Na_2_CO_3_ (75 g/L). The prepared solution was kept at 15 °C for 5 min. Absorbance was measured at 700 nm by an UV/Vis spectrophotometer (Synergy HT, BioTek Instruments Inc., Winooski, VT, USA). A calibration curve was prepared using a standard solution of gallic acid (2500–100 mg/L, *R*^2^ = 0.99), and the final values were expressed as milligrams of gallic acid equivalents (GAE) per gram of dry plant material (mg GAE/g dry plant). The total flavonoid content (TFC) was determined by aluminum chloride colorimetric method [24]. An aliquot (500 µL) of the plant extract was mixed with 2 mL of distilled water and 150 µL of NaNO_2_ solution (5%). After 6 min, 150 µL of AlCl_3_ solution (10%) was added and allowed to stand further 6 min; thereafter, 2 mL of NaOH solution (4%) and 200 µL of distilled water were added to the mixture. Then, the mixture was properly mixed and allowed to stand for 15 min, and the absorbance was measured at 510 nm by an UV/Vis spectrophotometer (Synergy HT, BioTek Instruments Inc., Winooski, VT, USA). A calibration curve (400–0 mg/L, *R*^2^ = 0.99) was prepared using (+)−Catechin, and the results were expressed as milligrams of catechin equivalents (CE) per gram of dry plant (mg CE/g dry plant).

#### 2.3.5. Identification and Quantification of Individual Phenolic Compounds

Individual phenolic compounds were analyzed by Shimadzu Nexera X2 UHPLC chromatograph equipped with Diode Array Detector (DAD) (Shimadzu, SPD-M20A, Kyoto, Japan) using a previously validated method, as described by Ferreira-Santos et al. [25]. Separation was performed on a reversed phase Acquity UPLC BEH C18 column (2.1 mm × 100 mm, 1.7 μm particle size; from Waters) and a precolumn of the same material at 40 °C. The flow rate was 0.4 mL/min. HPLC grade solvents water/formic acid 0.1% (A) and acetonitrile (B) were used. The elution gradient for solvent B was as follows: from 0 to 5.5 min eluent B at 5%, from 5.5 to 17 min linearly increasing from 5 to 60%, from 17 to 18.5 min linearly increasing from 60 to 100%; last, the column is equilibrated at 5% from 18.5 to 30 min. Phenolic compounds were identified by comparing their UV spectra and retention times with those of corresponding standards. Quantification was carried out using calibration curves for each pure phenolic compound standard, using concentrations between 250–2.5 mg/L, and the limit of detection (LOD) and limit of quantification (LOQ) were calculated for as previously reported by Busaranon et al. [26]. In all cases, the coefficient of linear correlation was *R*^2^ > 0.99 (Appendix A). Compounds were quantified and identified at different wavelengths (209–370 nm). All analyses were made in triplicate.

### 2.4. Bioactivities

#### 2.4.1. Antioxidant Activity

The antioxidant activity was measured using DPPH, ABTS and FRAP methods, to evaluate distinct mechanisms of action of the extracts. The free radical scavenging (DPPH) and the radical cation decolorization (ABTS) assays were conducted as described by Ballesteros et al. [27] with some modifications. Calibration curves were prepared with a standard solution of Trolox (250–15 μM, *R*^2^ = 0.998, for DPPH; and 800–31.25 μM, *R*^2^ = 0.996, for ABTS) and a corresponding control was used for each solvent. The radical scavenging activity for DPPH and ABTS methods (% inhibition) was calculated as Equation (4)

(4)
$$\%\;\mathrm{Inhibiton}\;=\;\frac{A_{c}-A_{s}}{A_{c}}\times 100$$

where *A_s_* is the sample absorbance and *A_c_* the control sample absorbance. The results were expressed as micromoles of Trolox equivalent (TE) per gram of dry plant (μmol TE/g dry plant).

The ferric reducing antioxidant power (FRAP) assay was performed as described by Meneses et al. [28]. A calibration curve was prepared using an aqueous solution of ferrous sulphate (800–100 μM, *R*^2^ = 0.98). FRAP values are expressed as micromoles of ferrous equivalent per g of dry plant (μmol Fe^2+^/g dry plant).

#### 2.4.2. Antimicrobial Activity

From all the extracts produced, three were selected for presenting distinctive results in terms of chemical profile (specifically, phenolic compounds content) and antioxidant activity. The bacteria tested were *Salmonella enterica* subsp. *enterica* serovar Typhimurium ATCC 43971, *Listeria monocytogenes* WDCM 00019, *Staphylococcus aureus* ATCC 6538, and *Escherichia coli* (clinical isolate), obtained from the Polytechnic Institute of Bragança stock collection. Bacteria strains were subcultured twice by streaking on blood agar and incubated at 37 °C for 48 h and then 24 h to ensure that bacterial cells were in the exponential growth phase. Following incubation in agar, single colonies from the second plate were inoculated into individual tubes containing sterile water and the bacterial suspensions were adjusted to a concentration of approximately 1.5 × 10^8^ CFU/mL (0.5 McFarland).

The minimal inhibitory concentrations (MIC) were determined by broth microdilution method according to the Clinical and Laboratory Standards Institute (CLSI) recommendations [29], with some modifications. The minimal bactericidal concentrations (MBC) were also determined by subcultivation of 10 μL of the microplate wells containing extracts at 20 mg/mL and 10 mg/mL into blood agar plates. The lowest concentration that showed no growth after this subculturing was regarded as the MBC.

The results were expressed in milligrams per milliliter of the resuspended lyophilized extracts (mg/mL). The MIC experiments were performed four times (*n* = 4) and the MBC tests were carried in duplicate (*n* = 2).

### 2.5. Statistical Analysis

Principal component analysis (PCA) was performed using the prcomp function from the factoextra package [30] to evaluate the contribution of variables (essays) and factors (plant, method, solvent) to the discrimination of extracts. Groupwise summary statistics (mean ± standard error) were calculated by plant, extraction method, and solvent, for each attribute (extraction yield, each chemical characterization assay and each antioxidant assay) using the summary_by function from the doBy package [31].

To assess the main effects of plant, extraction method, and solvent, and the interactions between those terms on each variable (essay), three-way interaction linear nonparametric models were built using the art function from the ARTool package [32], which applies an aligned rank transformation to every model. This transformation was done to enable a nonparametric analysis of variance (α = 0.05), as the normality assumptions were not met. The three-way interaction “plant×method×solvent” was included in the model to provide an adequate fit.

For each variable, pairwise comparisons of levels within single factors were conducted using the emmeans function from emmeans package [33], coupled with the artlm function from the ARTool package [32]. Superscript letters indicating significantly different values (*p* < 0.05) were defined according to the results of the emmeans function.

Statistical analysis was conducted in R software (version 3.6.2) [34].

## 3. Results and Discussion

With the potential to be used as food additives, basil, lemon balm, French lavender, sage, spearmint, and tarragon were used to produce twenty-four extracts, testing two extraction methods and two solvents per plant. The methods tested were solid-liquid and Soxhlet extractions because they generally offer good extraction results and are easy to implement, thus justifying their widespread use in the food industry to extract bioactive compounds [35]. Water and ethanol 70% (*v*/*v*) were selected as extraction solvents because herbal extracts should be produced using water, ethanol, or their binary mixtures, while toxic organic solvents should be avoided [6].

### 3.1. Influence of Extraction Yield, Chemical Characteristics, and Antioxidant Activity on Extracts Differentiation

To visualize the influence of extraction yield, chemical characteristics, and antioxidant activity on the differentiation of extracts, PCA was conducted (Figure 1).

The first two principal components, PC1 and PC2, accounted for most of the variance observed, 51.4% and 20.4%, respectively. While the first component, PC1, indicates that dissimilarities across the horizontal axis are mostly due to distinct phenolic contents and antioxidant activities; the second component, PC2, reflects the contribution of the photosynthetic pigments and extraction yield to the differentiation of samples.

In Figure 1, variables with little contribution to extracts distinction will appear closer to the plot origin, whereas variables with greater contribution will be further from the center of the plot. From this, the variables with the highest contribution to extracts differentiation are those associated with antioxidant activity (ABTS: 13.42%; DPPH: 12.95%; FRAP: 12.86%), TFC (12.20%), TPC (11.98%), and photosynthetic pigments (chlorophyll-a and -b, 13.09% and 9.90%, respectively). The variables with lower contribution were extraction yield, total protein content, and carbohydrate content (6.36%, 4.14% and 3.10%, respectively). These results reveal that extraction yield, total protein content, and carbohydrate content were fairly similar across the samples produced, but divergencies were mainly found in terms of phenolic contents, photosynthetic pigments, and antioxidant activities.

Figure 1 also provides insight on correlations between variables: positively associated variables will have approximately the same loading (i.e., distance from the plot origin) and will appear close to each other on the plot, whereas negatively correlated variables will appear diagonally opposite each other [36]. In this sense, the PCA shows that TPC and TFC are positively correlated with antioxidant activity (ABTS, DPPH and FRAP), an expected result due to the redox properties of phenolic compounds, which allow for adsorption and neutralization of free radicals, quenching of singlet and triplet oxygen, or decomposition of peroxides [37]. Several other studies on various plant materials have also reported on the strong correlation between phenolic compounds and antioxidant activity [38,39,40,41].

### 3.2. Influence of Extraction Method, Solvent, and Plant Type on Extracts Differentiation–Principal Component Analysis

Score plots of the first two components of the PCA were also produced to display the grouping of plant extracts by extraction method, solvent, and plant type, as shown in Figure 2A–C, respectively.

While the ellipses aim to group samples according to the method or solvent used, it is noticeable that three aqueous extracts are within the Et70 ellipse on Figure 2B. This is a result of three solid-liquid water extracts of lemon balm that revealed high chlorophyll contents, comparable to those of hydroethanolic extracts, unlike other aqueous extracts.

The score plots produced showed that the discrimination between extracts obtained from different solvents (Figure 2B) and different plants (Figure 2C) is greater than that attained between extraction methods (Figure 2A). This better discrimination arises from the greater difference in chemical characteristics and antioxidant properties among extracts obtained using different solvents or feedstocks than the different extraction methods. The influence of the solvent used on the extraction of phenolic compounds and antioxidant potential has also been reported by other research groups: Meneses et al. [28] pointed out the difference in antioxidant activity and total phenols content in brewer’s spent grains extracts when using water or organic solvents; Teofilović et al. [42] demonstrated the impact of different polarity solvents on the total phenolic and flavonoid contents of basil extracts; and Martins et al. [43] produced *L. tridentata* extracts with varying antioxidant activity, total phenols and flavonoids contents by using distinct extraction solvents.

Nevertheless, the chemical composition and antioxidant activity of extracts are also influenced by the extraction method used, even if such impact is less noticeable from Figure 2B. This effect of the extraction method on phytochemical constituents and antioxidant capacity was also reported by Scollard et al. [44] and Dhanani et al. [45].

Analyzing Figure 2C, the ellipses of spearmint, French lavender, and sage overlap, indicating similar phenolic contents and antioxidant activities (yet, different amounts of photosynthetic pigments, as revealed by the various heights of the ellipses, in PC2). However, they differentiate from the other three plants: lemon balm, basil, and tarragon. The figure suggests that lemon balm extracts contain the highest phenolic and flavonoid contents and antioxidant activity, as the extracts are in the same direction of the arrows of TPC, TFC, DPPH, ABTS and FRAP (see Figure 1). On the other hand, tarragon extracts contain the lowest quantity of phenolic compounds and most reduced antioxidant potential, as samples appear in the opposite direction.

### 3.3. Influence of Extraction Method, Solvent, and Plant Type on Extracts Differentiation–Main Effects and Interactions

In addition to the principal component analysis conducted, to further study the extracts differentiation and characteristics, groupwise summary statistics were calculated by plant, extraction method, and solvent, for each assay. Furthermore, three-way interaction models were built to assess the main effects of plant, extraction method, and solvent, and the interactions between those terms. These results are displayed in Table 1.

Despite the improved discrimination achieved due to solvent and plant types (Figure 2B,C) rather than between extraction methods (Figure 2A), the results of the main effects in Table 1 reveal the significant impact (*p* < 0.05) of all three terms on the extraction yield, chemical characteristics, and antioxidant capacities of the plant extracts.

In most assays, all three terms had a significance level of *p* < 0.001. The plant term, in particular, showed a significance level of *p* < 0.001 for all assays. The exceptions were found for the other two terms, method and solvent. Their impact on the TPC and carbohydrate assays appears to be less significant than that of plant type, therefore suggesting a greater difference in phenolic and carbohydrate contents between extracts obtained from distinct plants than from distinct methods or solvents—a result otherwise expected due to the specificities of each plant. The effect of plant type (*p* < 0.001) on total protein content and DPPH assays was also found to be stronger than the effect of solvent type (0.001 < *p* ≤ 0.01). The results of the DPPH and ABTS assays were the only ones found to be independent from one of the factors, namely the extraction method (0.05 < *p* ≤ 1), which is also indicated by the same superscript letter, in both columns.

The groupwise summary statistics in Table 1 provide information on the overall means and standard errors of each level of the main effects. From these statistics, it appears that the solid-liquid technique improves extraction yields and results in extracts with high levels of chlorophyll, carbohydrates and phenolic compounds (*p* < 0.05 for these assays), whereas Soxhlet extraction produces extracts with greater content in proteins and flavonoids, associated with a high reducing antioxidant power (determined by the FRAP test) (*p* < 0.05 for these assays). Evaluating the results by solvent type, water seems to be more effective (*p* < 0.05) in extracting proteins, phenolic compounds, and carbohydrates, whereas ethanol 70% (*v*/*v*) appears to be more efficient (*p* < 0.05) in chlorophylls and flavonoids recovery.

In terms of feedstock, lemon balm did not only exhibit higher and distinctive (*p* < 0.05) results, compared to the remaining plants, on TPC, TFC and antioxidant assays, but it also presented the highest values in extraction yield, chlorophyll-b, and carbohydrate contents (*p* < 0.05). Moreover, lemon balm also revealed the second highest values in total protein and chlorophyll-a content, although it not significantly different from the plant displaying the highest outcome in such assays (namely, French lavender presented the greatest total protein content, whereas tarragon showed the highest chlorophyll-a content). Such distinctive results (*p* < 0.05) in terms of flavonoids and total phenolic contents suggest the great potential of lemon balm to be used as a food preservative against oxidation and microbial spoilage because phenolic compounds have been associated with antioxidant and antimicrobial activities [5,46].

Interaction terms were also included in the models to provide information on whether the effect of one term depends on the level of one or more terms. In this sense, when they are statistically significant, it would not be correct to generalize the trends pointed out by the main effects without considering the interactions.

Apart from the interaction “method×solvent” in the case of chlorophyll-a and carbohydrate content (both 0.1 < *p* ≤ 1), and the interaction “plant×method” in the DPPH assay (*p* < 0.1), all two-way interactions were found significant (*p* < 0.05) for all assays. This means, for example, for the significant interactions “plant×solvent” and “plant×method” on TPC (both *p* < 0.001), that the effects of solvent and extraction method on the total phenolic content, respectively, are different for each plant. These interactions can be visualized in Figure 3A,B, respectively.

In Figure 3A, each violin plot displays the distribution of TPC values obtained from both extraction methods, for each solvent and plant combination tested. Similarly, in Figure 3B, each violin plot displays the distribution of TPC values obtained from both solvents, for each extraction method and plant combination tested. The height of the violin plot indicates the distribution of the values obtained, while the varying width indicates the frequency of data points in each region.

The interactions are indicated by the different slopes of the dashed lines, across the six plants. For example, from Figure 3A, it is clear that the effect of solvent type is very different for lemon balm than it is for sage. In case there was no significant interaction, the dashed lines of the various plants would be practically parallel to each other.

The significance of these interactions also indicates the distinct abilities of different solvents and extraction techniques in retrieving various compounds from the raw material and their influence on the antioxidant capacity of extracts (as discussed earlier, and according to results previously reported by other research groups [28,42,43,44,45]), for each plant type.

The “method × solvent” term reveals that, for the same extraction method, the results of each assay will depend on the solvent used. Moreover, it implies that, for the same solvent, the results of each assay will vary according to the extraction technique selected. This suggests that the four method/solvent combinations tested must be assessed to identify the one leading to the best or worst outcomes, in each assay (Appendix A).

In this sense, considering the results in Appendix A and adding to the discussion on the main effects, the solid-liquid technique does improve extraction yields and carbohydrates contents, particularly when the solvent is water, while high levels of chlorophylls and phenolic compounds can be achieved using ethanol 70% (*v*/*v*). On the other hand, Soxhlet extracts have the greatest content in proteins when using ethanol 70% (*v*/*v*), while the greatest flavonoid content and high reducing antioxidant power (FRAP) is obtained from aqueous extracts. Furthermore, water does seem to be more effective than ethanol 70% (*v*/*v*) in extracting proteins and phenolic compounds, when Soxhlet extraction is used, compared to the solid-liquid one. Oppositely, ethanol 70% (*v*/*v*) is more efficient than water in flavonoids recovery, but only when solid-liquid extraction is conducted. Nonetheless, these are overall results that vary depending on the plant selected, due to its specific characteristics.

Despite the nonsignificant main effects “method” and “solvent” on the results of the DPPH and ABTS assays, the significance of the “method × solvent” term in both cases (*p* < 0.001) reveals the existence of an interaction between these variables that affects the outcomes of such assays. These interactions on the outcomes of the DPPH and ABTS assays are shown by the distinct slopes of the dashed lines in Figure 4A,B, respectively).

From Appendix A, the highest values on the DPPH assay were largely derived from the aqueous Soxhlet extracts, followed by the hydroethanolic solid-liquid ones. The exceptions were sage hydroethanolic Soxhlet extract and basil and tarragon aqueous solid-liquid extracts. As for the ABTS assay, the combination leading to the highest overall values was solid-liquid extraction using ethanol 70% (*v*/*v*), and the second-highest was Soxhlet extraction with water as solvent.

Focusing on the results of the total phenolic contents in Appendix A, hydroethanolic Soxhlet extracts generally presented the lowest (or among the lowest) TPC, when contrasted with the other method/solvent combinations (aqueous and hydroethanolic solid-liquid extracts, and aqueous Soxhlet extracts). This result is likely a consequence of alcoholic solvents being generally very effective in extracting phenolic compounds, as they improve the solubility of such compounds from the raw material to the solvent medium [47,48,49]; however, because this study used a higher temperature (90 °C) during the Soxhlet extraction—compared to the one used for solid-liquid extraction (60 °C)—it may have promoted thermal degradation and oxidation of the compounds of the hydroethanolic Soxhlet extracts [48]. Chin et al. observed a similar outcome in tea, reporting a decrease in total polyphenols concentration from Soxhlet extraction carried out at 70 °C, compared to those obtained from maceration at 40 °C [50]. In this sense, it is possible to assume that, for Soxhlet extractions carried for seven recycles (3.5 to 4 h) at such high temperature, water may be the most appropriate solvent, whereas for solid-liquid extractions, either water or a mixture of water/ethanol is adequate. Otherwise, if Soxhlet extractions are carried out using ethanol as solvent, the appropriate extraction time and temperature must be assessed for optimum recovery of phenolic compounds from the plant matrix, as also suggested by Alara et al. [51].

The reported effect of the interaction “method × solvent” on the TPC can be visualized in Figure 5. In this sense, and given the results from Appendix A, the method/solvent combinations that could be selected for their potential in producing extracts of increased phenolic and flavonoid contents, and high antioxidant activities are Soxhlet extraction using water as solvent, and solid-liquid extraction using ethanol 70% (*v*/*v*). Between the two, the latter combination is highly promising for industrial applications as it is less time-consuming than Soxhlet extractions and it does not require any specific equipment.

### 3.4. Influence of Extraction Method and Solvent on Phenolic Profile of Plant Extracts

To further study the phenolic profile of the extracts, tentative identification and quantification of compounds was performed by UPLC. In total, fifteen compounds were identified in this study (Table 2).

The results showed that rosmarinic, ferulic and ellagic acids, naringin, hesperidin, resveratrol and quercetin were present in all plant extracts. They also showed that chlorogenic, vanillic, syringic, 3,4-dihydroxybenzoic, *o*-coumaric and ferulic acids, *p*-coumaric acid/epicatechin, kaempferol, resveratrol, and quercetin were undetected or found in concentrations below 250 mg/L extract, depending on the extract (Table 2). On the other hand, the compounds found in higher concentrations (between 267 and 1369 mg/L extract) were cinnamic, rosmarinic and ellagic acids, naringin and hesperidin.

Cinnamic acid was present at concentrations between 280–487 mg/L extract in the hydroethanolic solid-liquid extracts of spearmint, lemon balm and sage.

Rosmarinic acid was found at high concentrations in the aqueous (324–448 mg/L extract) and hydroethanolic (523–679 mg/L extract) Soxhlet extracts of spearmint, lemon balm and sage; in the aqueous solid-liquid extract of basil and tarragon (274 and 341 mg/L extract, respectively); and in the hydroethanolic solid-liquid extract of basil, spearmint, and tarragon (292–355 mg/L extract).

Ellagic acid showed high concentrations only in Soxhlet extracts, namely in the hydroethanolic extracts of tarragon, spearmint, and French lavender (416–554 mg/L extract) and the aqueous ones of tarragon, spearmint, lemon balm, basil, and sage (279–645 mg/L extract).

Naringin was present at high concentrations (267–894 mg/L extract) in both aqueous extracts of tarragon, in both Soxhlet extracts of sage, in the hydroethanolic solid-liquid extract of sage, and in the aqueous and hydroethanolic solid-liquid extract of basil and lemon balm, respectively.

Hesperidin was found in high amounts (279–996 mg/L extract) in all sage extracts, in spearmint and lemon balm Soxhlet extracts (both aqueous and hydroethanolic), and in the hydroethanolic Soxhlet and solid-liquid extracts of tarragon and French lavender, respectively.

Other researchers have also studied the phenolic profile of the plant materials used in our work. Nunes et al. performed the characterization of phenolic compounds from *L. stoechas* L. methanolic extracts [52]. Their research identified rosmarinic, ferulic, chlorogenic and vanillic acids, among other compounds. From our French lavender extracts, rosmarinic, ferulic and chlorogenic acids were also detected (the latter in only one of the extracts), while vanillic acid was never detected. Zgórka and Głowniak studied the phenolic profile of sage, basil and lemon balm extracts [53]. Their work indicated the presence of vanillic acid in sage and basil (approximately 25 and 6 μg/g dry plant, respectively). In our study, vanillic acid was detected in the aqueous solid-liquid sage extract at 250 μg/g dry plant (12.4 mg/L extract) and in two basil samples obtained by solid–liquid extraction at 260 and 340 μg/g dry plant (12.9 and 17.2 mg/L extract). Their research also revealed the existence of ferulic acid in sage (around 50 μg/g dry plant); and rosmarinic acid (the most predominant compound) in basil, lemon balm and sage (approximately 11650, 9690 and 5120 g/g dry plant, respectively). Our study also identified ferulic acid in all sage extracts, and rosmarinic acid in all basil, lemon balm and sage extracts. Zgórka and Głowniak did not identify chlorogenic acid in any of the tested plant extracts, which was also the case in our study, depending on the extraction method and solvent used [53].

Kivilompolo et al. also performed the characterization of phenolic acids from sage, basil and spearmint extracts [54]. Their research identified rosmarinic acid in basil, spearmint and sage (3080, 5620 and 9960 μg/g dry plant)—like we did in our study—as well as chlorogenic acid in basil; vanillic acid in sage and spearmint; and syringic, *p*-coumaric and ferulic acids in all herb extracts. With some exceptions, most of these outcomes agree with those presented in Table 2. In contrast, Kivilompolo et al. [54] reported the presence of vanillic acid in basil (140 μg/g dry plant) and chlorogenic acid in sage and spearmint (230 and 310 μg/g dry plant). In our study, vanillic acid was only detected in two basil extracts, as previously referred; chlorogenic acid was never identified in sage, and only detected in one spearmint extract (180 μg/g dry plant; 8.77 mg/L extract).

Slimestad et al. investigated the phenolic profile of tarragon extracts, reporting a limited number of phenolic compounds in this herb, of which chlorogenic acid stands out as one of the main constituents (1607 μg/g dry plant) [55]. In another study, by Mumivand et al. [56], HPLC analysis of twelve tarragon extracts (from different origins) indicated that chlorogenic acid (5.73 to 37.07 μg/g dry plant) and syringic acid (3.17 to 29.01 μg/g dry plant) were present in all extracts, and that such compounds were generally found in higher quantities (except in a few samples). Quercetin, vanillic, ferulic and *p*-coumaric acids were also identified, but not in all extracts, and usually in lower amounts (also, with some exceptions).

Overall, our results agree with the findings of Slimestad et al. and Mumivand et al. [55,56] in the sense that the compounds reported in those studies were also identified in some of our extracts, except for syringic acid. Another difference is that chlorogenic acid was not the predominant compound in our tarragon extracts (in fact, it was only found in one of them).

The work of Mumivand et al. highlights that, for the same plant species, the origin of the plant has an impact on the phenolic profile of the extracts, hence the variability of the results [56]. For this reason, it is not unexpected to observe discrepancies among outcomes of different studies, even for the same plant species, as seen here in some cases.

In this sense, it should be stated that comparison of results is important but must be done carefully because the outcomes are dependent on various factors, including the plant characteristics. Climate, cultivation method, stage of development of the plant and time of harvesting, etc., are likely to influence the phenolic composition of the extract produced [54].

In addition to the impact of plant specificities, to visualize the influence of extraction methods and solvents on the phenolic profile of extracts, principal component analysis was carried out (Figure 6A,B, respectively).

The results suggest a greater difference in phenolic composition for extracts obtained from different methods than from different solvents. Moreover, the results indicate that Soxhlet extracts generally contain higher concentrations of rosmarinic acid, ellagic acid, hesperidin, and kaempferol. In contrast, solid-liquid extracts present greater amounts of cinnamic and *o*-coumaric acids, resveratrol, and quercetin, for example.

Despite the less noticeable discrimination between the phenolic compounds profile of aqueous and hydroethanolic extracts, it is perceptible that higher concentrations of rosmarinic acid, resveratrol, and hesperidin could be associated with hydroethanolic extracts, which is explained by the poor solubility of these compounds in water [49].

With this knowledge on the phenolic profile of the plant extracts, it is possible to have insight on the best method/solvent/feedstock combination that should be selected depending on the desired phenolic compound to be retrieved. Furthermore, our work allows simultaneous comparison between the plants studied both in terms of chemical composition and some biological potentialities, which were discussed in the previous sections, thus highlighting the promising antioxidant activity of some of these extracts.

### 3.5. Antimicrobial Activity

After considering the outcomes of the principal component analyses and the distinctive results in terms of chemical profile (more specifically, phenolic content) and antioxidant activity of some extracts, three were selected for the determination of MIC and MBC against four pathogens. The extracts chosen were all hydroethanolic, produced by solid-liquid extraction, using spearmint, sage, and lemon balm. The results obtained are displayed in Table 3.

The MIC is defined as the lowest concentration of an antimicrobial agent that completely inhibits growth of the organism in the microdilution wells as detected by the unaided eye [29]. In this study, all extracts examined showed promising results, with sage extract revealing the greatest potential. In fact, the sage extract resulting from hydroethanolic solid-liquid extraction produced the lowest MIC, 0.625 mg/mL, against *S. aureus*. This extract also revealed greater inhibitory action against *S. enterica* ser. Typhimurium, and equivalent or higher inhibitory action against *L. monocytogenes* and *E. coli*, compared to the remaining extracts.

While Gram-negative bacteria are generally more resistant to natural extracts than Gram-positive bacteria, with some exceptions [57], this was only evident in the case of *S. enterica* ser. Typhimurium, which revealed to be the least susceptible bacterium regardless of the bioactive extract. *E. coli*, on the other hand, did not reveal greater resistance to the plant extracts than the Gram-positive bacteria tested.

All plant extracts failed to kill the bacteria at the tested concentrations of 20 mg/mL and 10 mg/mL. Therefore, the MBC of these extracts were greater than 20 mg/mL.

In a study by Btissam et al. [58], hydroethanolic sage extracts obtained through maceration showed generally lower inhibitory effects than those determined for our extracts, with MIC values of 3.12 mg/mL for *L. monocytogenes*, 1.56 mg/mL for *S. aureus*, 25 mg/mL for *S. enterica* and 50 mg/mL for *E. coli*. Contrarily to our work, in this study, determination of the MBC was possible, with values ranging from 3.12 to 100 mg/mL. In another work, by Stanojević et al. [59], aqueous sage extracts obtained by maceration revealed MIC values of 20 and 40 mg/mL for *S. aureus* and *E. coli*, respectively.

Scherer et al. [60] tested the antibacterial activities of spearmint extracts obtained by maceration with methanol, acetone, and dichloromethane against strains of *E. coli* and *S. aureus*; however, none of the extracts revealed significant antimicrobial activity. Another study, by Caleja et al. [61], analyzed the antimicrobial activity of spearmint aqueous infusions, which presented higher inhibitory activity than the extracts of this study: the MIC values were 0.5 mg/mL for *L. monocytogenes*, 0.25 mg/mL for *S. typhimurium*, and 0.5 mg/mL for *E. coli* (MBC values of 1, 0.5 and 1, respectively).

The same work by Caleja et al. also determined the antibacterial activity of lemon balm aqueous infusions against *L. monocytogenes*, *S. typhimurium* and *E. coli*: all strains were equally affected by the extract, with MIC and MBC values of 1 mg/mL and 2 mg/mL, respectively [61]. These results indicate higher antimicrobial capacity of such infusion than that of our extracts. Oppositely, another study, by Ceyhan et al. [62], also testing the inhibitory effects of lemon balm, revealed MIC values against *S. aureus* of 3.12 and 6.25 mg/mL for the aqueous and hydroethanolic extracts, respectively; and MIC values against *E. coli* O157:H7 of 50 and 6.25 mg/mL for the aqueous and hydroethanolic extracts, respectively.

The large variety of MIC values described in literature for a single plant material may be explained by the impact of multiple factors on the chemical profile of the extracts (extraction method, solvent, and plant specificities, as mentioned before). Moreover, despite the existence of a standard protocol for antimicrobial susceptibility testing, the results may also vary depending on the bacterial strain selected, for example.

The potential of plant extracts as antimicrobials in food products is also dependent on the matrix selected. In this sense, it is crucial to perform experimental trials in food matrices to attest the functionality of the extract because the bioactivities determined in vivo will most likely be different that those found in vitro, a possible consequence of interactions between the plant extract and the food components and properties. Additionally, tests that can determine appropriate doses in foods, dose-response effects, and therapeutic dosages must be conducted, and the cost of usage of plant extracts must be assessed (to validate its commercial potential), as well as the impact on the sensory attributes of the food product (appearance, aroma, taste, texture, etc.).

## 4. Conclusions

The outcomes of this study provide insight on the phytochemical profile, antioxidant activity and antimicrobial potential of various plant extracts; they also provide insight on the effect of edible plant type, extraction methods and solvents on such characteristics.

The results show that both extraction method and solvent have an impact on most of the chemical characteristics and antioxidant profile of the extracts. Nonetheless, a greater difference is observed between extracts obtained using distinct solvents than distinct methods. The results also indicate the existence of interactions between the factors plant type, extraction method and solvent, affecting some of the chemical and antioxidant characteristics of extracts.

Overall, hydroethanolic solid-liquid extracts showed great potential as biopreservatives, due to their high phenolic contents, antioxidant activities and antimicrobial capabilities. Lemon balm, spearmint, and sage extracts presented the highest phenolic and flavonoid contents and strong antioxidant activities. Additionally, they revealed antimicrobial activity against four important foodborne pathogens (*S*. *enterica* ser. Typhimurium, *E. coli*, *L. monocytogenes* and *S. aureus*). These outcomes support the potential of lemon balm, spearmint, and sage extracts to be incorporated in foods as preservatives against oxidation and microbial spoilage. Nevertheless, further trials must be carried out to attest the functionality of these extracts, which is likely influenced by the food matrix and their cost of usage (to validate its commercial potential), as well as tests that can determine, for instance, appropriate doses in foods, dose-response effects, and therapeutic dosages.

## Figures and Tables

**Figure 1 foods-10-00673-f001:**
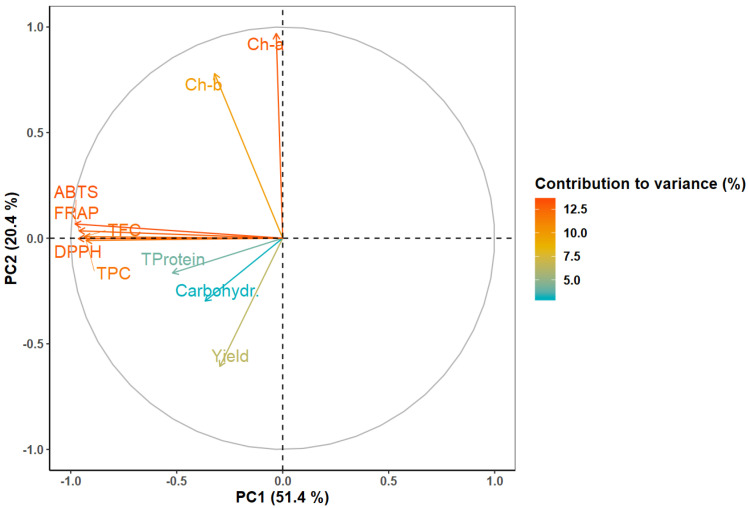
Loading plot of the first two components of the principal component analysis (PCA).

**Figure 2 foods-10-00673-f002:**
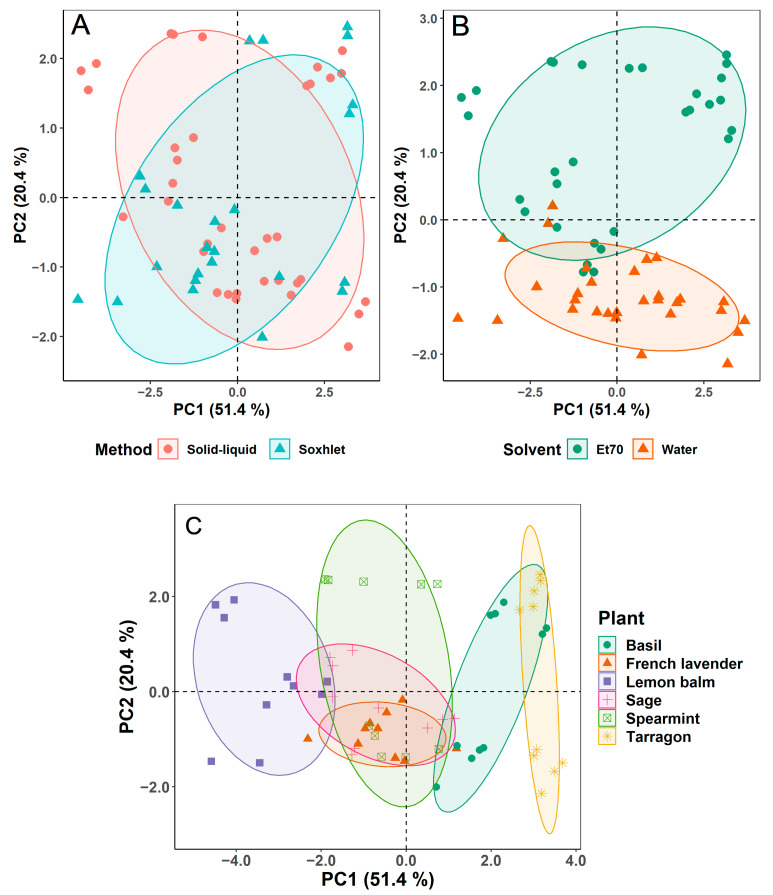
Score plots of the first two components of the principal component analysis (PCA) grouped by extraction method (**A**), solvent (**B**) and plant type (**C**).

**Figure 3 foods-10-00673-f003:**
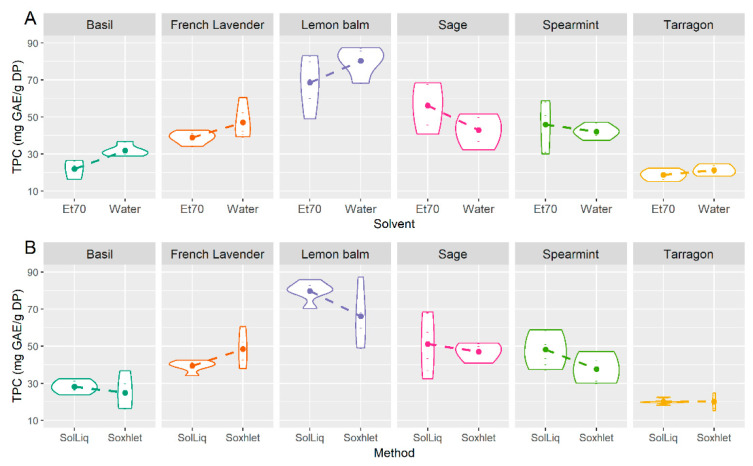
Interaction plots “plant × solvent” (**A**) and “plant × method” (**B**) on the total phenolic content of plant extracts.

**Figure 4 foods-10-00673-f004:**
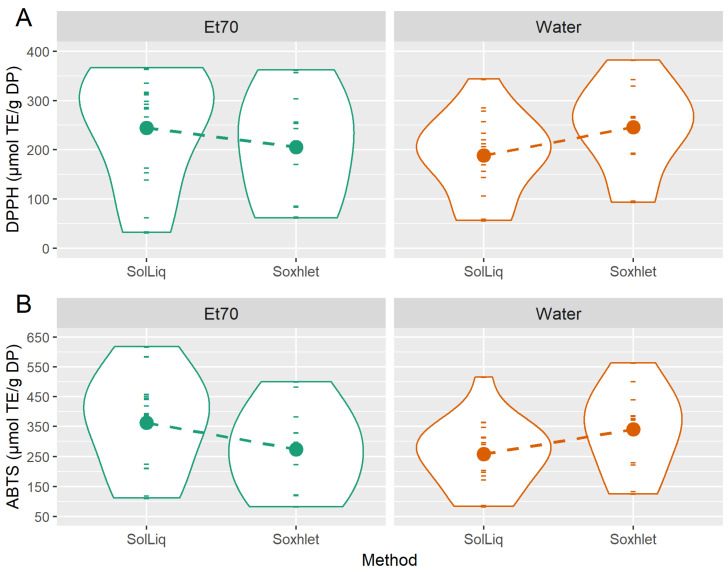
Interaction plots “method × solvent” on the outcomes of the DPPH (**A**) and ABTS (**B**) assays of plant extracts.

**Figure 5 foods-10-00673-f005:**
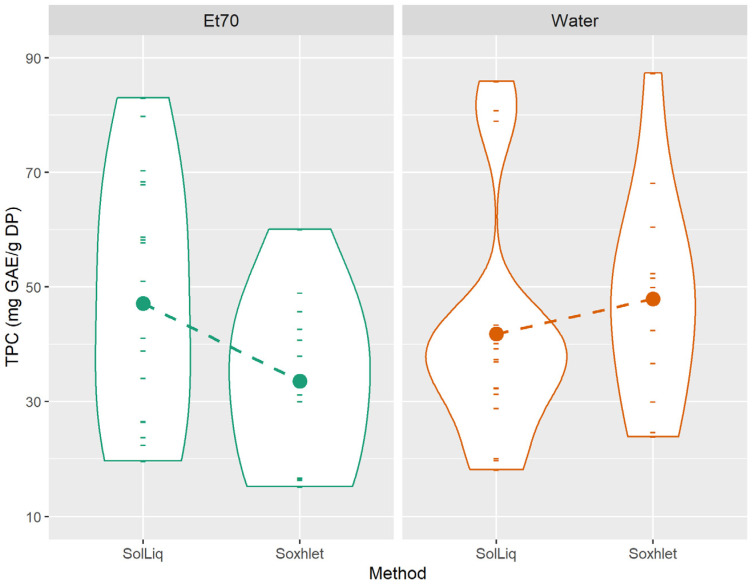
Interaction plot “method × solvent” on the total phenolic content of plant extracts.

**Figure 6 foods-10-00673-f006:**
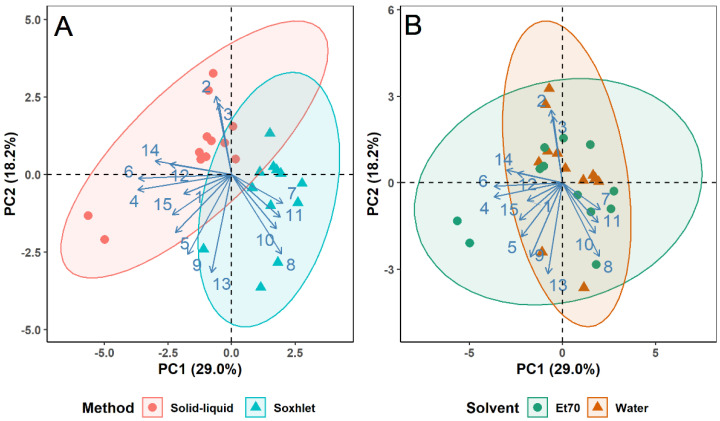
Score plots of the first two components of the principal component analysis (PCA) of phenolic compounds grouped by extraction method (**A**) and solvent (**B**). Phenolic compounds: 1–Chlorogenic acid, 2-Vanillic acid, 3-Syringic acid, 4-Cinnamic acid, 5-*p*-coumaric acid + epicatechin, 6-*o*-coumaric acid, 7-Rosmarinic acid, 8-Ellagic acid, 9–Naringin, 10–Hesperidin, 11–Kaempferol, 12–Resveratrol, 13-Ferulic acid, 14–Quercetin, 15-3,4-Dihydroxybenzoic acid.

**Table 1 foods-10-00673-t001:** Groupwise summary statistics (mean ± standard error) by plant, method, and solvent, for each chemical characterization and antioxidant assay, and significance of the main effects and interactions of the models.

	Yield (%)	Ch-a (μg/g DP)	Ch-b (μg/g DP)	TProtein (μg BSAE/g DP)	TFC (mg CE/g DP)	TPC (mg GAE/g DP)	Carbohydr. (μg GE/g DP)	DPPH (μmol TE/g DP)	ABTS (μmol TE/g DP)	FRAP (μmol Fe^2+^/g DP)
Plant										
Tarragon	23.1 ± 1.34 ^bc^	96.9 ± 28.7 ^a^	132 ± 33.0 ^bc^	4.19 ± 0.83 ^c^	8.05 ± 0.78 ^c^	20.0 ± 0.95 ^e^	17.6 ± 2.44 ^b^	61.8 ± 6.59 ^c^	107 ± 6.07 ^c^	191 ± 11.5 ^e^
Spearmint	21.5 ± 1.33 ^d^	92.6 ± 23.8 ^a^	149 ± 35.0 ^ab^	8.91 ± 0.81 ^b^	30.5 ± 1.23 ^b^	44.0 ± 3.16 ^c^	13.7 ± 1.39 ^cd^	259 ± 14.4 ^b^	361 ± 20.4 ^b^	722 ± 31.0 ^b^
Lemon balm	26.2 ± 1.24 ^a^	96.1 ± 14.8 ^a^	209 ± 33.1 ^a^	10.4 ± 0.77 ^a^	45.6 ± 4.86 ^a^	74.4 ± 3.90 ^a^	22.0 ± 1.76 ^a^	345 ± 11.0 ^a^	507 ± 28.6 ^a^	1013 ± 75.5 ^a^
Basil	22.2 ± 1.87 ^cd^	68.9 ± 16.8 ^b^	108 ± 22.1 ^d^	6.02 ± 0.94 ^c^	16.2 ± 1.07 ^c^	26.9 ± 2.07 ^d^	11.8 ± 1.21 ^d^	149 ± 12.0 ^c^	194 ± 12.6 ^c^	376 ± 28.0 ^d^
French lavender	25.2 ± 0.78 ^ab^	43.5 ± 6.60 ^c^	115 ± 9.18 ^cd^	10.7 ± 1.54 ^a^	32.2 ± 0.94 ^b^	43.1 ± 2.45 ^c^	21.6 ± 0.88 ^a^	241 ± 20.5 ^b^	326 ± 25.4 ^b^	614 ± 51.4 ^c^
Sage	22.4 ± 0.41 ^cd^	59.8 ± 12.4 ^bc^	99.9 ± 13.4 ^d^	8.98 ± 1.16 ^b^	30.9 ± 2.18 ^b^	49.5 ± 3.87 ^b^	16.5 ± 1.65 ^bc^	265 ± 13.8 ^b^	358 ± 23.1 ^b^	752 ± 44.5 ^b^
Method										
Solid-liquid	24.8 ± 0.55 ^a^	80.8 ± 9.85 ^a^	163 ± 16.6 ^a^	6.15 ± 0.46 ^b^	26.3 ± 2.39 ^b^	44.5 ± 3.46 ^a^	18.0 ± 0.92 ^a^	216 ± 17.0 ^a^	310 ± 24.6 ^a^	579 ± 47.8 ^b^
Soxhlet	21.4 ± 0.91 ^b^	69.7 ± 12.7 ^b^	93.6 ± 8.46 ^b^	11.3 ± 0.71 ^a^	28.6 ± 2.83 ^a^	40.7 ± 3.68 ^b^	16.1 ± 1.44 ^b^	226 ± 20.5 ^a^	307 ± 28.7 ^a^	660 ± 65.7 ^a^
Solvent										
Water	21.1 ± 0.73 ^a^	28.1 ± 4.41 ^b^	86.6 ± 12.2 ^b^	8.43 ± 0.70 ^a^	25.5 ± 2.23 ^b^	44.3 ± 3.54 ^a^	18.1 ± 1.29 ^a^	212 ± 16.1 ^a^	291 ± 23.2 ^b^	577 ± 49.4 ^b^
EtOH 70%	25.8 ± 0.50 ^b^	124 ± 8.06 ^a^	184 ± 14.5 ^a^	7.97 ± 0.75 ^b^	29.0 ± 2.88 ^a^	41.7 ± 3.67 ^b^	16.3 ± 0.95 ^b^	229 ± 20.6 ^b^	327 ± 29.0 ^a^	646 ± 60.2 ^a^
Main effects										
Plant	***	***	***	***	***	***	***	***	***	***
Method	***	***	***	***	***	**	*	.	NS	***
Solvent	***	***	***	**	***	*	*	**	***	***
Interactions										
Plant × Method	**	***	***	***	***	***	***	.	*	*
Plant × Solvent	***	***	***	***	***	***	**	***	***	***
Method × Solvent	***	NS	***	***	***	***	NS	***	***	***
Plant × Method × Solvent	*	**	***	***	***	**	.	*	*	**

DP: dry plant; Mean values with different superscript letters in a column are significantly different. “NS”: *p* < 1; “.”: *p* < 0.1; “*”: *p* < 0.05; “**”: *p* < 0.01; “***”: *p* < 0.001.

**Table 2 foods-10-00673-t002:** Identification and quantification of phenolic compounds present in the extracts produced.

Phenolic Compound (mg/L Extract)			Chlorogenic Acid	Vanillic Acid	Syringic Acid	Cinnamic Acid	*p*-Coumaric Acid + Epicatechin	*o*-Coumaric Acid	Rosmarinic Acid	Ellagic Acid	Naringin	Hesperidin	Kaempferol	Resveratrol	Ferulic Acid	Quercetin	3,4HBA
Soxhlet	H_2_O	Tarragon	27.4 ± 0.79	nd	nd	9.61 ± 1.11	165 ± 10.0	62.6 ± 6.38	45.9 ± 2.60	645 ± 31.0	270 ± 18.9	99.4 ± 7.57	nd	15.5 ± 2.74	111 ± 2.83	3.31 ± 0.64	9.16 ± 0.56
		Spearmint	8.77 ± 0.22	nd	nd	nd	nd	nd	324 ± 32.4	279 ± 21.9	55.7 ± 9.32	561 ± 45.9	nd	25.7 ± 2.12	55.1 ± 1.24	5.21 ± 0.26	nd
		Lemon balm	12.2 ± 1.76	nd	nd	nd	nd	nd	448 ± 109	373 ± 179	105 ± 31.1	901 ± 232	nd	59.3 ± 21.8	18.2 ± 6.79	12.6 ± 5.26	nd
		Basil	nd	nd	nd	nd	12.3 ± 1.72	nd	128 ± 2.73	420 ± 126	69.6 ± 3.30	206 ± 1.53	nd	5.94 ± 5.52	51.6 ± 2.89	4.13 ± 0.07	nd
		French lavender	nd	nd	nd	nd	33.4 ± 12.1	nd	198 ± 0.75	75.9 ± 5.61	71.8 ± 10.8	85.0 ± 13.5	93.7 ± 6.69	103 ± 6.35	74.3 ± 12.1	10.3 ± 2.01	nd
		Sage	nd	nd	nd	nd	184 ± 16.6	nd	435 ± 41.8	587 ± 423	523 ± 33.8	900 ± 71.4	nd	4.92 ± 0.62	161 ± 11.7	5.53 ± 0.04	nd
	EtOH 70%	Tarragon	nd	nd	nd	nd	93.3 ± 0.75	2.45 ± 0.25	38.9 ± 1.44	472 ± 41.6	133 ± 8.66	61.4 ± 1.37	nd	16.3 ± 0.42	68.6 ± 2.55	5.79 ± 2.04	1.27 ± 0.14
		Spearmint	nd	nd	nd	nd	nd	nd	555 ± 30.7	416 ± 32.4	92.7 ± 9.53	1131 ± 63.2	63.2 ± 2.70	68.1 ± 5.23	49.1 ± 4.78	19.4 ± 1.87	nd
		Lemon balm	nd	nd	nd	nd	nd	nd	679 ± 61.8	238 ± 0.48	93.5 ± 2.66	1369 ± 105	63 ± 3.37	97.9 ± 7.57	0.81 ± 0.06	8.60 ± 0.65	nd
		Basil	nd	nd	nd	nd	nd	nd	143 ± 5.48	150 ± 5.07	34.3 ± 0.72	242 ± 10.4	nd	10.8 ± 1.21	9.00 ± 0.05	6.39 ± 2.55	nd
		French lavender	nd	nd	nd	nd	nd	nd	244 ± 27.9	554 ± 69.2	116 ± 0.61	495 ± 45.5	77.1 ± 8.35	127 ± 7.71	88.8 ± 10.1	27.2 ± 6.52	nd
		Sage	nd	nd	nd	nd	4.14 ± 0.84	nd	523 ± 3.85	249 ± 11.0	537 ± 8.05	996 ± 113	98.8 ± 4.43	44.5 ± 5.01	163 ± 3.85	33.0 ± 4.47	nd
Solid-liquid	H_2_O	Tarragon	nd	nd	nd	25.0 ± 3.32	57.6 ± 4.27	44.2 ± 5.08	341 ± 28.8	42.9 ± 5.33	267 ± 32.5	43.3 ± 6.59	nd	33.6 ± 1.21	34.9 ± 2.51	15.9 ± 1.01	nd
		Spearmint	nd	nd	nd	150 ± 3.79	53.7 ± 6.40	20.5 ± 0.14	204 ± 53.2	28.3 ± 2.58	22.8 ± 0.53	110 ± 26.6	nd	75.2 ± 8.25	42.8 ± 1.28	71.4 ± 10.9	nd
		Lemon balm	nd	nd	nd	56.3 ± 8.94	nd	99.9 ± 59.9	129 ± 11.9	82.1 ± 3.43	116 ± 5.43	31.6 ± 0.14	nd	90.9 ± 8.22	80.7 ± 1.71	43.1 ± 0.81	nd
		Basil	nd	17.2 ± 0.30	10.5 ± 0.24	80.5 ± 1.32	103 ± 11.8	33.9 ± 1.35	274 ± 10.9	40.2 ± 3.54	287 ± 1.17	144 ± 9.77	nd	46.4 ± 0.95	39.0 ± 0.17	34.2 ± 0.62	nd
		French lavender	nd	nd	nd	80.5 ± 6.93	nd	30.9 ± 6.91	120 ± 42.9	39.2 ± 0.32	63.5 ± 27.0	116 ± 5.51	nd	95.2 ± 2.01	64.5 ± 11.1	29.5 ± 0.39	nd
		Sage	nd	12.4 ± 0.05	9.33 ± 0.63	53.3 ± 1.26	nd	36.7 ± 0.79	173 ± 101	43.3 ± 14.5	78.0 ± 12.0	279 ± 30.8	nd	93.9 ± 3.53	35.9 ± 3.55	54.6 ± 7.35	nd
	EtOH 70%	Tarragon	nd	9.92 ± 0.03	nd	37.2 ± 4.66	nd	33.2 ± 4.99	355 ± 5.15	38.5 ± 7.58	123 ± 0.52	292 ± 5.89	2.73 ± 0.38	95.6 ± 2.38	47.0 ± 8.11	33.7 ± 0.29	nd
		Spearmint	nd	nd	nd	280 ± 10.6	69.0 ± 3.17	27.8 ± 0.61	333 ± 57.3	28.6 ± 0.40	62.9 ± 2.16	223 ± 12.7	1.13 ± 0.32	111 ± 7.50	60.7 ± 13.9	55.7 ± 2.67	nd
		Lemon balm	71.2 ± 0.48	nd	nd	487 ± 15.8	123 ± 1.36	111 ± 92.4	185 ± 27.2	50.5 ± 4.25	894 ± 51.8	3.71 ± 3.34	nd	126 ± 9.33	108 ± 35.9	41.2 ± 0.11	nd
		Basil	64.9 ± 1.25	12.9 ± 0.18	nd	79.9 ± 4.15	84.4 ± 10.7	25.5 ± 2.01	292 ± 4.31	31.0 ± 2.62	65.1 ± 7.86	188 ± 13.5	nd	85.4 ± 1.27	50.2 ± 7.47	18.1 ± 0.21	nd
		French lavender	32.2 ± 0.22	nd	nd	121 ± 2.21	68.5 ± 0.85	31.6 ± 0.59	127 ± 18.4	49.9 ± 14.7	77.8 ± 2.31	123 ± 8.73	nd	96.2 ± 0.86	55.1 ± 0.56	27.3 ± 0.03	nd
		Sage	nd	nd	nd	485 ± 66.3	119 ± 3.22	94.9 ± 4.13	170 ± 13.6	52.2 ± 9.61	279 ± 16.1	805 ± 40.0	nd	200 ± 13.0	78.5 ± 4.35	129 ± 4.76	16.0 ± 0.30

3,4HBA: 3,4-Dihydroxybenzoic acid; nd: not detected.

**Table 3 foods-10-00673-t003:** Minimal inhibitory concentration (mg/mL) of hydroethanolic solid-liquid extracts obtained from spearmint, sage, and lemon balm, against *L. monocytogenes*, *S. aureus*, *S. enterica* ser. Typhimurium, and *E. coli*.

Plant	*L. monocytogenes*	*S. aureus*	*S.* Typhimurium	*E. coli*
Sage	2.5–5	0.625	10	1.25
Spearmint	2.5	1.25	20	1.25
Lemon balm	5	2.5	20	2.5

## Data Availability

Summary data available upon request.

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
