# Peer review of "Chemical Profile and Bioactivities of Extracts from Edible Plants Readily Available in Portugal"

_foods, 2021, doi:10.3390/foods10030673_

Round 1

Reviewer 1 Report

My comments are following:

  1. The aim should be better defined in the abstract.
  2. The following reference should be included since it is describing the possibility to use essential oils: Simona, J., Dani, D., Petr, S., Marcela, N., Jakub, T., & Bohuslava, T. (2021). Edible films from carrageenan/orange essential oil/trehalose—structure, optical properties, and antimicrobial activity. Polymers13(3), 332.
  3. The extraction with the Soxlet apparatus included what kind of solvent?
  4. Bradford assay has to be described more in detail.
  5. The determination of carbohydrate content has to be described more in detail.
  6. The determination of chlorophyll content has to be described more in detail.
  7. The determination of total flavonoid content has to be described more in detail.
  8. Why anova test was not performed, or it is not described in the material and methods section?
  9. The section explaining artool package is not included. Figure 3 should be described more clearly, same as Figure 4.
  10. The statistical analysis is not included in Table 2.

Reviewer 2 Report

The reviewed manuscript is a revised version of a previous submission. I am happy that the authors have decided to resubmit the manusript after the improving of the suggested points.

Author Response

The authors appreciate the reviewer's comments that helped improving the manuscript.

Reviewer 3 Report

The paper was improved according Reviewers' suggestions.

Author Response

(The authors gave the same response as above.)

Round 2

Reviewer 1 Report

I still believe that the inclusion of ANOVA test would increase the quality of the work. 

Author Response

This manuscript is a resubmission of an earlier submission. The following is a list of the peer review reports and author responses from that submission.

Round 1

Reviewer 1 Report

The manuscript entitled “Chemical profile and bioactivities of extracts from edible 2 plants readily available in Portugal”, authored by Silva and colleagues, deals with the evaluation of rosemary, lemon balm, basil, tarragon, sage, and spearmint as potential food additives. Analysis included the profile of phytochemical composition via two different extraction methods (Soxhlet and maceration). The authors performed several statistical analyses to discuss and give strength to the obtained results. The manuscript is innovative, and contains very interesting data, but completly miss of a discussion section.

Moreover, the introduction should be supplemented with further information concerning the plant materials used in the study. Indeed, they are only covertly mentioned in this section, and should be better discussed as they are the protagonists of this manuscript.

The section related to identification and quantification of bioactive compounds performed via HPLC system miss in the description of chromatographic condition used for the analysis. Moreover, validation analysis also are not completely described. In particular, the authors should provide information regarding LOD (limit of detection), LOQ (limit of quantification) and ME (Matrix Effect) values.

Authors should move the table containing the quantification of bioactive compounds from supplementary file in the manuscript, since it contains important data for this study. The authors should compare their identifications and quantifications with those previously reported in the literature.

A great limitation of the article consists in the almost complete absence of a discussion. The authors do not compare the obtained data with those previously published, but limit the manuscript to a simple comment of the statistical data. I strongly encourage the authors to include a section related to the discussion of the data.

Reviewer 2 Report

The paper of Silva et al. deals with the bioactive compounds deriving from plants. It seems to be very interesting while the usage of synthetic preservatives should be limited in the food industry. Results are well discussed; numerous valuable thesis were propounded. However, I have some doubts that I would like to see answered:

  1. The presence of bioactive compounds (cinnamic acid, rosemaric acid etc.) was confirmed using chromatographic methods. In my opinion, additional experiments should be performed, e.g. MS for qualitative analysis.
  2. Authors claimed that isolated compounds revealed antimicrobial activity. However, MIC value are quite high (around few mg/mL), and MBC was greater than 20 mg/L. Do the compounds have potential to be commercially used? What is the yield of isolation, and therefore the total cost of the usage such extracts (which may change the taste in such high concentration). I realize this is a far-reaching question, however Authors claimed about the importance of these investigations and their potential usage.

Reviewer 3 Report

My comments are following:

  1. The whole abstract is not formatted in the same way.
  2. Chapters 2.3.1., 2.3.3., 2.3.4.: the methods are not described in detail.
  3. Chapter 2.3.5: the sample preparation is not described, mobile phase and gradient system are not described too.
  4. Chapter 2.4.2.: the conditions of cultivation are not described.
  5. Table 1.: it is not clear for which extractions are the results shown, PLANT.
  6. Table S1 and S2: there is no statistical analysis.
  7. Table 2.: there are no standard deviations and statistics.
  8. There is a lack of references in the results and discussion part.
  9. There is no capital DISCUSSION.